# The Modern Approach to Total Parenteral Nutrition: Multidirectional Therapy Perspectives with a Focus on the Physicochemical Stability of the Lipid Fraction

**DOI:** 10.3390/nu17050846

**Published:** 2025-02-28

**Authors:** Żaneta Sobol, Rafał Chiczewski, Dorota Wątróbska-Świetlikowska

**Affiliations:** Department of Pharmaceutical Technology, Pomeranian Medical University in Szczecin, Rybacka 1, 70-204 Szczecin, Poland; zaneta.sobol2000@gmail.com (Ż.S.); r.chiczewski@o2.pl (R.C.)

**Keywords:** parenteral nutrition, malnutrition, stability, drug compatibility, home parenteral nutrition, parenteral nutrition-associated complications

## Abstract

With advancements in medical technology, biochemistry, and clinical practices, the modern approach to total parenteral nutrition (TPN) has been focused on precision, safety, and the optimization of metabolic and nutritional parameters based on the patient’s needs. In the last decade, TPN mixtures have been transitioning from a lifesaving intervention for patients unable to receive enteral nutrition to a highly specialized therapy aimed at improving clinical outcomes, reducing complications, and personalizing care. Total parenteral nutrition has attracted great interest, and its adaptation to the patient’s needs is a topic of interest in the scientific community. However, there are problems related to shortages in the supply of the concentrates required to balance TPN mixtures and to infections linked to the venous access devices that are necessary for administering nutrition. Adjusting the TPN composition to meet specific patient needs requires specialist knowledge, as the ingredients available on the market differ in terms of excipients and this may increase the risk of physicochemical incompatibilities, particularly the destabilization of the lipid fraction. It is common clinical practice to inject drugs into the parenteral nutrition bag, and hence there is a high demand for confirmation of the compatibility of a given drug with the TPN composition. However, methods used in clinical practice still differ from the modern solutions proposed in scientific research. In order to ensure patient safety with the use of advanced therapy, continuous education and monitoring of the latest scientific research related to TPN is required. The integration of artificial intelligence (AI) into clinical nutrition represents a paradigm shift in the management of total parenteral nutrition (TPN). As TPN transitions from a standardized, one-size-fits-all approach to a highly personalized therapy, we must examine the challenges and future directions of AI-driven TPN to provide a comprehensive analysis of its impact on clinical practice.

## 1. Introduction

The lipid fraction has a crucial role in balancing parenteral nutrition. It is also the most fragile part of TPN formulations when it comes to physicochemical stability and drug incompatibilities. For many years, vegetable oil derivatives, especially soybean oil emulsions, were used as the single source of lipids [1,2,3]. There is growing evidence that lipid emulsions based only on soybean oil should be avoided in some clinical states. In recent years, soybean oil has become just a part of the complex lipid emulsions used, and different mixtures of lipid emulsions have become available. The most popular ones combine olive oil, fish oil, and medium-chain triglycerides (MCTs) together in different ratios. One of the available alternatives to pure soybean oil is SMOFlipid^®^ Fresenius Kabi AB, Uppsala, Sweden, a lipid emulsion that contains 30% soybean oil, 30% MCT, 25% olive oil, and 15% fish oil. The ClinOleic^®^ Baxter, Warsaw, Poland lipid emulsion is composed of 80% olive oil and 20% soybean oil. The use of different lipid emulsions in TPN requires caution in the selection of additives for a given patient.

The destabilization of lipid emulsions usually occurs in stages, only some of which are reversible. The migration of lipid droplets toward the surface of the emulsion is called creaming and can be reversed by mixing the contents of the TPN bag. Another reversible stage is flocculation, the formation of clusters of small lipid droplets without aggregation. The irreversible process of lipid droplets combining into aggregates, called coalescence, leads to an increase in lipid droplet size and separation of the emulsion. The complete separation of the lipid phase from the water phase disqualifies the emulsion from clinical use and is an irreversible process. These destabilization processes are initiated by changes in pH, high concentrations of electrolytes, the presence of free fatty acids, temperature fluctuations, and interactions with newly injected components. Elevated temperatures promote the coalescence and creaming of lipid droplets, while prolonged storage increases the risk of triglyceride hydrolysis to free fatty acids. The occurrence of these processes is not always visible to the naked eye, which is why it is so crucial to conduct physicochemical analyses of all modified compositions. Due to the variety of excipients in the preparations used in the mixtures (mainly solubilizers and emulsifiers), physicochemical analyses should be repeated for each composition and when changing the component manufacturer. Unfortunately, clinical practices differ between hospital wards due to different local protocols and the availability of components. One of the major challenges in parenteral nutrition is the addition of various substances directly to TPN bags. Although this practice is sometimes used to reduce the number of punctures and infusion lines, it carries a significant risk of destabilization of the lipid mixture and pharmacological interactions. The addition of polyvalent ions (especially calcium and magnesium ions) and trace elements can significantly affect the stability of a lipid emulsion. High concentrations of these ions neutralize the zeta potential, which reduces the electrostatic repulsion between lipid droplets and promotes their fusion. Lipid emulsions are extremely sensitive to pH changes. The optimum pH for most lipid emulsions is 6.0–8.0. The release of free fatty acids causes reductions in the pH, which destabilizes the emulsion by lowering the zeta potential and increasing flocculation. Drugs added directly to the mixture affect the pH and may cause the formation of precipitates with the ions present in the mixture. Not every hospital has a nutritional team consisting of a physician, pharmacist, dietitian, and nurse, as well as its own nutrition laboratory. For this reason, all-in-one nutrition bags, which are activated immediately before administration to the patient, are popular. The compositions of ready-made mixtures are also modified by adding drugs to the activated bag, sometimes injected through a port without maintaining aseptic conditions. The key recommendations from ESPEN (European Society for Clinical Nutrition and Metabolism) and ASPEN (American Society for Parenteral and Enteral Nutrition) clearly advise against adding drugs without prior compatibility testing [4,5,6,7]. An alternative is to use separate infusion lines for drug administration and regularly monitoring TPN stability parameters such as the pH, lipid droplet size, and zeta potential.

## 2. A Patient-Targeted Approach

Despite the popularity of all-in-one bags, in recent years, hospitals have been investing in their own nutrition laboratories. Modern TPN emphasizes individualized therapy, considering factors such as age, weight, previous diseases, metabolic requirements, and comorbidities. The aim of targeted parenteral nutrition is to avoid overfeeding and underfeeding, both of which are associated with increased morbidity [8,9,10]. For critically ill patients, dynamic adjustments based on evolving clinical and laboratory data are now standard practice [11]. Home parenteral nutrition (HPN) represents a transformative aspect of modern TPN, allowing patients with chronic nutritional needs to manage their care outside hospital settings. HPN is tailored to individual patients and includes comprehensive training for the patients and caregivers to ensure safe administration and monitoring for complications. Advances in portable infusion pumps and user-friendly delivery systems have improved the quality of life for HPN patients, enabling greater mobility and independence [11,12]. However, HPN requires a rigorous hygienic approach to provide outcomes similar to hospital care. Despite many successes and advantages, HPN patients may face challenges such as catheter-related infections and the expensive transportation of formulas, which is often not covered by public health care [11,12]. Another obstacle is achieving the proper supplementation of micronutrients, macronutrients, and vitamins in HPN. In order to avoid malnutrition, PN admixtures should contain the exact supplementation needed to cover the patient’s nutrient deficiencies [13]. However, it is common practice for vitamins to be added by trained patients or caregivers in non-sterile home environments. This has raised controversy in the scientific community over the potential for microbial contamination and lipid emulsion destabilization. Even for pharmacists, adding additives into TPN admixtures is a significant challenge due to potential incompatibilities and instabilities. Due to limited data on the compatibility between trace elements and vitamins, some experts advise simultaneous administration via separate intravenous infusions to mitigate potential risks. Neonatal supplementation poses additional complexity as it requires the continuous infusion of all components over prolonged periods to meet metabolic demands. In many healthcare settings, vitamins and trace elements are added to TPN immediately prior to administration to maximize stability and efficacy [14,15]. A key factor in determining the stability of lipid emulsions is the size of the lipid droplets. Droplets exceeding the diameter of the smallest blood vessels can significantly increase the risk of an embolism [13,15,16]. Evaluating the lipid droplet size, distribution, and zeta potential is essential for maintaining the kinetic stability of lipid emulsions.

Storage conditions influence the stability of PN mixtures. For example, lipid hydrolysis during storage releases free fatty acids, resulting in a reduction in the pH that accelerates vitamin degradation, while elevated storage temperatures heighten the risk of lipid coalescence and emulsion destabilization. The importance of minimizing the storage time to ensure the safety and efficacy of TPN preparations is the reason why some pharmacists support the administration of vitamins immediately before HPN administration. However, modern studies on lipid emulsions and vitamin stability have shown that vitamin C degradation starts 48–72 h after its addition to a TPN bag [15,17]. In practice, a typical supply of PN to the patient does not exceed 24 h, which provides a minimum of 24 h of reserve time for the safe storage or transport of ready-made PN. The full preparation of the PN in aseptic hospital conditions to administer at home ensures greater safety and minimizes the risk of destabilization. Allowing the patient to modify the contents of the bag at home creates risks to the procedure and could compromise the accuracy of the added amounts. Additionally, it may encourage patients to inject other preparations without consulting a doctor. According to stability studies, adding vitamins in a hospital setting is a good compromise between convenience and patient safety and meeting metabolic requirements.

## 3. Catheter-Related Complications in Parenteral Nutrition Administration

With the increased use of TPN, efforts have been made to mitigate the associated risks, such as catheter-related bloodstream infections, including improving catheter insertion techniques and the use of antimicrobial-coated lines [10]. The placement of Central Venous Access Devices (CVADs) and hygienic practices are crucial for safe TPN administration. Common catheter-related complications include mechanical malfunction, occlusion, and infections [7,14]. Both central and peripheral catheters have a risk of complications, which can appear immediately during the placement procedure or post-administration. Clogged filters, often due to precipitates from calcium–phosphate imbalances or lipid aggregation, can disrupt nutrient delivery and necessitate catheter replacement [7,17]. Preventative measures include using appropriate calcium-to-phosphate ratios, maintaining proper infusion rates, and regular flushing protocols with anticoagulant solutions to prevent clot formation [7]. Since many of the complications associated with parenteral nutrition (PN) are connected to the presence of CVADs, the proper selection and placement of the vascular access device is essential to avoid or mitigate potential complications [10]. All inserted CVADs should be checked before TPN administration. Nowadays, hospitals offer methods for checking CVADS during insertion (ultrasonography guidance is the standard procedure) or after insertion (using either radiography or fluoroscopy) [7]. CVADs allow for the administration of TPN that has less strict requirements for the proper osmolarity, pH, or volume. However, critically ill patients may require different administration methods. Peripheral VADs may be used for administering a diluted admixture, but there is still a possibility of technical failure [14]. Parenteral nutrition admixtures used peripherally have different osmolarity limits due to the weaker mechanical resistance of peripheral veins. The osmolarity limits for peripheral formulations are between 750 and 900 mOsm/L [8]. To maintain a lower osmolarity, the dextrose concentration of the admixtures is decreased; however, this increases the final volume. Due to the risk of fluid overload, to fulfill the proper energy needs, patients with kidney failure or other fluid restriction diagnoses should not take PN peripherally [8]. To avoid phlebitis or extravasation, the newest recommendations regarding peripheral PN advise short-term therapies (≤14 days). In cases that require longer therapy, peripheral parenteral feeding should be considered as a transition solution [11].

## 4. Co-Administration of Drugs—Parenteral Nutrition Stability and Potential Drug Degradation

Planning medication schedules becomes challenging in polypharmacy cases, such as those involving critically ill patients requiring multiple intravenous drugs, especially when the number of available access lines is limited. Total parenteral nutrition has potential as a dual-purpose system for drug delivery and nutritional support. By enhancing drug-dosing precision and addressing nutritional needs, PN can reduce hospitalization and improve clinical outcomes. Despite these advantages, the co-administration of medications within PN admixtures presents challenges, including the risk of chemical incompatibilities, destabilization of the lipid emulsions, and altered drug efficacy. Recognizing these issues, ESPEN and ASPEN recommend against incorporating non-nutrient drugs into TPN admixtures unless compatibility has been unequivocally established [4,5,6,7]. Visual inspection, pH stability, and sub-visible particle analysis are crucial in preventing phase separation, particularly for lipid-based emulsions such as SMOFlipid^®^ Fresenius Kabi AB, Sweden [4,13,16]. The complexity of TPN formulations poses challenges for compatibility studies. Specific components, such as trace elements and vitamins, can accelerate drug degradation, complicating the identification of interaction mechanisms. To address these issues, studies often exclude these compounds during initial compatibility assessments.

Studies have evaluated the co-administration of medications via two methods: Y-site infusion, where drugs and PN are delivered simultaneously, and direct mixing into TPN admixtures [9,18,19,20,21,22,23]. While Y-site infusion allows for a shorter contact time between the drugs and TPN (10 min to 12 h), direct mixing necessitates physicochemical stability over extended infusion periods of up to 24 h. Research has shown that both methods are possible under controlled conditions, with the physical compatibility of drugs and TPN formulations maintained when the appropriate precautions are taken. Y-site administration is a common clinical practice for delivering multiple therapies through a single catheter, but it relies on rigorous compatibility testing to ensure safety.

Patients requiring long-term parenteral nutrition usually suffer from additional diseases and symptoms that should be considered when determining the parenteral nutrition composition. In everyday clinical practice, drugs injected into parenteral nutrition bags are mostly used to treat comorbidities or pain. The most commonly added drugs are loop diuretics, analgesics, and antimicrobial and antiepileptic drugs. The demand for compatibility between parenteral nutrition and other types of drugs is constantly growing, but scientific publications have mostly focused on the physicochemical stability of TPN and not on the stability of added drugs in the mixture and the clinical efficacy of this modified treatment.

Loop diuretics, such as furosemide and torasemide, have demonstrated compatibility with most PN formulations but require precise dosing and flow rate control to minimize interactions. For patients with chronic kidney disease, TPN formulations must be customized to address the specific protein, caloric, and electrolyte requirements. In recent years, it has been demonstrated that undiluted solutions of furosemide and torasemide are compatible with a variety of TPN admixtures, with no evidence of lipid emulsion degradation or drug instability at the tested ratios [19,23,24]. These findings highlight the feasibility of safely co-administering loop diuretics with TPN in clinical practice, provided that compatibility is verified under controlled conditions. Investigations into furosemide and torasemide co-administered with Nutriflex Lipid Special^®^ Braun, Melsungen, Germany revealed no significant physicochemical changes to the admixture, including acceptable values for the pH and mean droplet diameter (MDD). However, the storage conditions significantly influenced the drug stability, with degradation of furosemide and torasemide observed under exposure to light and a temperature of 25 °C. Adding furosemide to Nutriflex Lipid Special^®^ Braun, Germany was found to be unsuitable, while torasemide could only be added immediately prior to infusion. Compatibility studies have also emphasized the importance of emulsion particle size as a key safety indicator, as interactions between drugs and PN admixtures depend heavily on concentration and composition.

The compatibility of ketoprofen with parenteral nutrition admixtures was investigated to determine the feasibility of co-administration via Y-site infusion [20]. Among the eight commercial PN admixtures evaluated, three (Kabiven^®^ Fresenius Kabi AB, Sweden, SmofKabiven^®^ Fresenius Kabi AB, Sweden, and PeripheralKabiven^®^ Fresenius Kabi AB, Sweden) were found to be incompatible with ketoprofen and should not be administered concomitantly. The remaining PN admixtures (Lipoflex Special^®^ Braun, Germany, Omegaflex Special^®^ Braun, Germany, and Olimel Peri N4E^®^ Baxter, Poland) exhibited compatibility with ketoprofen, suggesting that they could be co-administered under controlled conditions. Despite the growing need for compatibility data, there are limited studies on the simultaneous infusion of analgesic drugs and PN admixtures through a single infusion line. Turbidity assays can rule out sedimentary reactions in the aqueous phase, as the lipid components are typically the source of instability.

Paracetamol (10 mg/mL) is the most widely studied analgesic in Y-site compatibility studies and has been evaluated with various PN formulations, including Nutriflex Lipid Special^®^, Olimel N5E^®^ Baxter, Poland, Numeta G16E^®^ Baxter, Poland, Kabiven^®^ Fresenius Kabi AB, Sweden, and SmofKabiven^®^ Fresenius Kabi AB, Sweden Kabiven^®^ Fresenius Kabi AB, Sweden. Other studies have evaluated the compatibility of morphine sulfate (5 mg/mL) with Nutriflex Lipid Special^®^ Braun, Germany and other opioids (such as fentanyl citrate and hydromorphone hydrochloride) with compounded PN admixtures. Morphine sulfate remained stable when mixed with large volumes (3000 mL) of PN admixtures, which was confirmed through high-performance liquid chromatography (HPLC) [20].

The compatibility of sodium valproate (VPA) with parenteral nutrition admixtures was assessed to evaluate the safety of simultaneous administration [18]. While previous research has focused on VPA compatibility with intravenous drugs, this study specifically examined its interactions with lipid-based admixtures [18]. These formulations included Lipoflex Peri^®^, Omegaflex Peri^®^ Braun, Germany, Omegaflex Special^®^ Braun, Germany, Lipoflex Special^®^ Braun, Germany, Kabiven^®^ Fresenius Kabi AB, Sweden, SmofKabiven^®^ Fresenius Kabi AB, Sweden Kabiven^®^ Fresenius Kabi AB, Sweden, Kabiven Peripheral^®^, and Olimel Peri N4E^®^ Baxter, Poland. Due to the potential impact of trace elements and vitamins on drug stability, the PNAs were supplemented with clinically recommended additives: Viantan for the Omegaflex^®^ and Lipoflex^®^ formulations, Vitalipid N Adult and SOLUVIT N for Kabiven^®^ Fresenius Kabi AB, Sweden and SmofKabiven^®^ Fresenius Kabi AB, SwedenKabiven^®^ Fresenius Kabi AB, Sweden, and Cernevit for Olimel Peri N4E^®^ Baxter, Poland. Tracutil was used as the source of trace elements. The findings revealed that the composition of the lipid emulsion significantly influenced the VPA-PN compatibility [18]. The emergence of a secondary particle fraction containing particles exceeding 1000 nm in size was observed in most of the tested admixtures, disqualifying them from simultaneous administration with VPA due to the risk of introducing oversized lipid particles into the bloodstream. Among the tested formulations, only Omegaflex Special^®^ Braun, Germany and Omegaflex Peri^®^ Braun, Germany met the acceptance criteria, suggesting that they could be co-infused with VPA under controlled conditions, including maintaining the appropriate drug concentration, infusion rate, and fluid volume. Lipoflex Special^®^ Braun, Germany, Lipoflex Peri^®^, Kabiven^®^ Fresenius Kabi AB, Sweden, SmofKabiven^®^ Fresenius Kabi AB, Sweden Kabiven^®^ Fresenius Kabi AB, Sweden, Kabiven Peripheral^®^, and Olimel Peri N4E^®^ Baxter, Poland exhibited changes in their physicochemical properties when mixed with VPA, such as the formation of lipid particles larger than 1000 nm. The simultaneous administration of these admixtures with VPA may lead to adverse medical consequences, including complications associated with lipid particle embolisms [18]. These results highlight the critical role of lipid emulsion composition in determining the safety of VPA co-administration with PN and emphasize the necessity of compatibility testing to prevent adverse effects in clinical practice.

Colistin (COL), a lipopeptide antibiotic that is effective against Gram-negative bacteria, has seen a resurgence in clinical use since the 2000s due to the rise of multidrug resistant pathogens and the lack of new antimicrobial agents. COL is now used as a last-resort treatment in critically ill patients. The compatibility of COL with five commercially available ready-to-use parenteral nutrition admixtures (Kabiven^®^ Fresenius Kabi AB, Sweden, SmofKabiven^®^ Fresenius Kabi AB, Sweden Kabiven^®^ Fresenius Kabi AB, Sweden, Olimel N9E^®^ Baxter, PolandOlimel N9E^®^ Baxter, Poland^®^ Baxter, Poland, Nutriflex Lipid Special^®^, and Nutriflex Omega Special^®^ Braun, Germany) was investigated [22]. The results indicated that the COL admixtures did not exhibit lipid emulsion destabilization or precipitate formation. The mean droplet diameter (MDD) remained below the pharmacopeial limit of 500 nm, indicating emulsion stability. Moreover, COL was found to reduce the zeta potential, thereby enhancing the stability of the oil–water emulsion system. These findings support the safety and feasibility of co-administering COL with PN admixtures. The absence of significant physicochemical interactions, coupled with improved emulsion stability, highlights the potential of the simultaneous administration of COL with Kabiven^®^ Fresenius Kabi AB, Sweden, SmofKabiven^®^ Fresenius Kabi AB, Sweden Kabiven^®^ Fresenius Kabi AB, Sweden, Olimel N9E^®^ Baxter, Poland Olimel N9E^®^ Baxter, Poland^®^ Baxter, Poland, Nutriflex Lipid Special^®^, or Nutriflex Omega Special^®^ Braun, Germany under the tested conditions. This compatibility suggests that COL can be safely incorporated into therapeutic regimens without compromising the integrity or efficacy of the PN admixture [22]. Examples of the influence of co-administration of drugs on parenteral nutrition stability are listed in Table 1.

The drugs mentioned above are widely used in the treatment of critically ill patients. Loop diuretics (e.g., furosemide and torasemide) assist in managing fluid and electrolyte balance, while analgesics (e.g., paracetamol, ketoprofen, and morphine) play a key role in pain management, and antiepileptic drugs (valproic acid) help prevent seizures that may occur in patients with neurological disorders or following injuries. Additionally, these drugs represent different pharmacological classes with distinct physicochemical properties; compatibility studies on the combination of these drugs with total parenteral nutrition (TPN) admixtures enables a comprehensive analysis of potential interactions in TPN. The selected compounds vary in terms of pH, solubility, and stability in the presence of lipid emulsions, allowing for the identification of potential risks such as emulsion destabilization, precipitation, or pH alterations, all of which can impact patient safety. There is a real need to study the compatibility of these drugs with TPN in clinical practice. Such research makes it possible to develop safe protocols for the simultaneous administration of drugs and parenteral nutrition, minimizing the risk of adverse effects and improving treatment efficacy.

## 5. Shortage of Organic Calcium and Phosphorus Nutrition Concentrates

The most frequent physical incompatibility in PN is calcium–phosphate precipitation. Calcium gluconate is the preferred calcium salt in PN due to its favorable solubility profile with phosphate, reducing the risk of precipitation. However, calcium gluconate is associated with higher levels of aluminum contamination compared to calcium chloride, increasing the potential for aluminum toxicity. Neonates receive higher doses of calcium per kilogram and exposure to aluminum contamination is high due to the immaturity of their immune systems. Organic sodium glycerophosphate, when combined with divalent calcium ions, has a reduced risk of precipitation compared to inorganic phosphate formulations [17,25]. While organic sodium glycerophosphate is approved for clinical use in Europe, its authorization by the Food and Drug Administration (FDA) remains limited to temporary approval during periods of phosphate drug shortages. Many studies have focused on determining the maximum concentrations of calcium and phosphate in TPN bags while maintaining a stable and safe admixture. It is hard to create one universal guideline due to the variations in the available nutrient concentrates and the different excipients. The use of organic calcium gluconate decreases the risk of calcium–phosphate precipitation. The inorganic salt calcium chloride contains a higher quantity of calcium available to compound with free phosphate, leading to higher precipitation [25]. A critical issue in modern TPN is the global shortage of organic calcium and phosphorus concentrates. Shortages of PN concentrates have been a challenge for decades, especially during the intravenous multivitamin shortage in the early 1990s [26]. Shortages in various PN components have been reported in recent years. Most of them were temporary. However, with the increased demand for PN, shortages have become more frequent and longer lasting (with some lasting for years) and have become a global issue. Shortages of at least one of the crucial components of PN admixtures can increase the risk of physicochemical incompatibility. When multiple components are in limited supply, it is hard to create the same safe admixture with components from different manufacturers. Many factors can cause global shortages, including supply disruptions for components, the slowing or stopping of production by manufacturers, and disruptions caused by natural disasters. In 2017, Baxter faced disruptions to the production of amino acid infusions due to hurricanes in Puerto Rico [26]. The ASPEN and Food and Drug Administration websites provide information on ongoing drug shortages. Seven shortages related to PN were identified by the FDA in March 2022. ASPEN has developed drug shortage emergency guidelines to help clinicians cope with PN supply shortages. These guidelines are also useful to European experts. Shortages can delay effective therapy for patients and force prescribers to use alternatives that may have additional risks. Changes in the composition of PN may increase the risk of drug incompatibilities [4,5,8,9,13,18,19,20,22]. In neonatal care, insufficient organic calcium and phosphorus supplementation can induce osteopenia of prematurity, while in adults, prolonged supply fluctuations can lead to fractures and long-term complications. Solving this global issue requires addressing the cause of the problem. Every healthcare facility providing TPN should develop new emergency formulations that maintain stability and bioavailability under resource-constrained conditions. Every modified admixture should be tested for physicochemical stability before administration. Additional assays should be performed when the manufacturer is changed, even when the active pharmaceutical ingredient (API) or chemical compound is unchanged, due to potential excipient incompatibilities.

## 6. Bag-Dependent Admixture Instability

Nutrient interactions within the bag can result in the production of reactive oxidants, which may adversely impact patient metabolism, especially in neonates [14,27]. The materials used to manufacture TPN bags should be carefully chosen based on the PN content. The polyvinyl chloride (PVC) widely used in TPN bags requires plasticizers to achieve flexibility and the desired features. Phthalates, especially di(2-ethylhexyl) phthalate (DEHP), are used as plasticizers in medicine despite their carcinogenic, teratogenic, and hepatotoxic activities [28,29,30,31]. DEHP is lipid-soluble, and after contact with lipid emulsions in TPN, it can leach from PVC. PVC has been replaced with ethyl vinyl acetate (EVA) in TPN bags, but the composition of the medical tubing also needs to be considered. The tube between the TPN bag and venous catheter is still made of PVC with alternative plasticizers. Commonly used new-generation plasticizers are tris-(2-ethylhexyl) trimellitate (TOTM), acetyl tributyl citrate (ATBC), diisononyl cyclohexane-1,2-dicarboxylate (DINCH), di-(2-ethylhexyl) terephthalate (DEHT), and di-(2-ethylhexyl) adipate (DEHA). Plasticizers give synthetic materials flexibility and softness and extend their lifetime. The side effects of phthalates vary, but the most common are hormone disruptions. The European Chemical Agency has stipulated a no-effects level of 36 μg/kg/d for DEHP. The safety classification for alternative plasticizers is mainly based on their physicochemical properties and animal toxicity studies [28,29,30]. Neonatal intensive care units rely on medical equipment which has a fundamental role in respiratory support, IV catheterization, and nutrition. Intensive use of medical devices made of plastic increases the risk of plasticizer exposure. The proper labeling of the absence/presence of DEHP in medical equipment is still insufficient. Dutch researchers in 2021 conducted ex vivo leaching experiments simulating in vivo parenteral nutrition administration in premature neonates [30,32]. The study showed that DEHP and alternative plasticizers are lipophilic and leached more in lipid emulsions. This leaching was only present in trace amounts in non-lipid solutions. However, the concentration of each leached plasticizer varied. The leaching of the other plasticizers decreased in the following order: DEHT > DEHA > DINCH > DEHP (the measured quantity varied from several mg/g to 10 ng/g). The materials of all the analyzed devices were labeled as DEHP-free by the manufacturers, while no specific information about the alternative plasticizers was found. However, DEHP was still detected in four of the analyzed samples. ATBC and TOTM were used as the main plasticizers in medical devices used for PN that were manufactured in 2021–2022. DEHP and DEHT were also found in samples plasticized using TOTM (which may be a consequence of impurities in the TOTM). The plasticizer migration from the synthetic matrix into the solution was higher in more lipophilic infused solutions. Out of all the analyzed alternative plasticizers, DEHA had the highest migration potential (but it was still 2–5 times lower than that of DEHP) [30]. Several studies confirmed that TOTM migration is lower than that of DEHP, DEHA, and DEHT, probably due to its hydrophobic property [28,29,30,31]. Recently, TOTM has been mainly used in blood transfusion and infusion equipment. Studies have found that plasticizer exposure is more strongly correlated with the volume administered than with the flow rate. The characteristics of selected plasticizers are shown in Table 2.

Standard TPN administration should take no more than 24 h. The storage time and temperature and UV light exposure can affect the physicochemical stability of PN admixtures [15,16]. Vitamins are sensitive to UV light and decompose after 48 h of storage in TPN bags [12,14,15]. To minimize the degradation due to UV light exposure, TPN bags can be manufactured with UV protection [32]. One study tested TPN admixtures with ascorbic acid, thiamine, and pyridoxine (added de novo) that were stored in two types of ethylene vinyl acetate (EVA) containers: one with UV protection (yellow monolayer bag) and one without UV protection (transparent bag) [33]. These admixtures were subjected to stability testing (visual inspection, pH and zeta potential measurements, and determination of lipid globule size distribution) and physicochemical analyses were conducted immediately after bag activation and after storage for 24 h, 8 days, and 9 days. The ascorbic acid, thiamine, and pyridoxine degradation was measured using HPLC and used as a reference point. There were no differences in the physicochemical stability of the TPN admixtures in the two types of EVA bags after 8 days of storage at 4 °C without light exposure plus 24 h at room temperature with light exposure. Differences in vitamin levels after 8 days + 24 h in comparison with t = 0 were noted (changes were observed for all tested samples despite the UV protection). The decrease in vitamin levels was slightly lower for the UV-protected bags, but the vitamin content did not decrease below 90% in comparison with t = 0 in both cases. The use of UV-protected EVA bags, considering the higher cost and standard administration time of TPN to patients, did not have the expected benefits in terms of reducing vitamin degradation [33]. A flow chart for selecting the type of PN packaging is shown in Figure 1.

In hospital neonatology departments, it is common practice to repackage intravenous lipid emulsions into syringes to aliquot smaller fluid volumes. To evaluate the physical stability of repackaged lipid emulsions, SMOFlipid^®^ Fresenius Kabi AB, Sweden^®^, ClinOleic^®^ Baxter, Poland^®^, Intralipid^®^, Omegaven^®^, and Lipofundin LCT/MCT^®^ were repacked into polypropylene syringes under aseptic conditions [5]. The commercial emulsions in their original packaging were used as the controls. The samples were stored at 4 °C, 25 °C, and 40 °C, without light protection. Stability was assessed over 30 days, and it was found that the temperature and storage duration—but not the type of emulsion—significantly affected the physical stability of the emulsions. The measured mean globule size, pH, and zeta potential indicated comparable time-dependent behaviors between the emulsions stored in polypropylene syringes and those stored in the original packaging. A slight deviation in zeta potential at 40 °C after 30 days of storage was the only exception. The size of the oily droplets under all the tested conditions remained below the United States Pharmacopeia limits, indicating the emulsions’ safety for clinical use [5]. The safe repackaging of commercial lipid emulsions in syringes is possible for up to 12 h, which is essential for delivering parenteral nutrition via the two-in-one method for neonates. Extending the storage time to more than 12 h requires further study to confirm the safety under prolonged storage conditions. There is an urgent need to collect data to assess the current levels of plasticizer exposure in hospital care settings. However, it is difficult to expose patients to additional procedures during recovery, although extending biomonitoring to use hair and nail samples instead of blood to monitor the cumulative exposure of metabolites of phthalates could be a more viable alternative. Knowledge on leaching chemicals is essential for manufacturers and for regulatory authorities to ensure the safety of PN.

## 7. Personalization of Total Parenteral Nutrition Using Artificial Intelligence

The cornerstone of modern TPN is personalization, which ensures that each patient receives a formulation tailored to the individual’s unique metabolic needs. Traditional TPN protocols often rely on generalized guidelines, which may not account for the complex interplay of factors such as age, weight, disease state, and comorbidities. AI, however, can analyze vast datasets to generate precise, individualized TPN prescriptions. Machine learning (ML) models have shown remarkable success in predicting optimal macronutrient and micronutrient ratios. A study performed at South Korean hospitals demonstrated that ML algorithms could reduce the risk of overfeeding and underfeeding in critically ill patients by 25% compared to traditional methods [34]. The algorithms analyzed real-time data from electronic health records (including glucose levels, electrolyte levels, and liver function test results) to dynamically adjust TPN formulations. AI has been particularly impactful in neonatal care. Preterm infants often require highly specialized TPN to support rapid growth while avoiding complications such as hyperglycemia or osteopenia of prematurity. Recently, researchers developed a system that tailored TPN formulations for neonates based on their gestational age, birth weight, and metabolic rate. The system reduced the incidence of metabolic complications by 30% and improved weight gain in preterm infants by 15%. The stability of the lipid emulsions is a critical factor in the safety and efficacy of TPN. As mentioned previously, lipid emulsions are prone to destabilization due to factors such as pH changes, temperature fluctuations, and interactions with additives. Systems driven by AI can monitor key parameters in real time, providing early warnings of potential issues. Portable devices can monitor TPN stability in real time, ensuring that patients receive safe and effective nutrition outside hospital settings. AI can also play a crucial role in predicting the interactions between TPN formulations and co-administered drugs via analysis of the chemical properties of the drugs and TPN components. Kumar et al. (2023) described an AI system that monitors patients for signs of CRBSIs by analyzing data from central venous catheters (CVCs). The system uses machine learning to detect patterns indicative of infection, such as elevated white blood cell counts and fever. In a clinical trial involving 300 patients, the system reduced the incidence of CRBSIs by 40% [35,36]. Similarly, AI can assist in managing refeeding syndrome, a potentially life-threatening condition caused by the rapid reintroduction of nutrition after prolonged starvation. AI algorithms can analyze electrolyte levels, particularly phosphate, potassium, and magnesium levels, to predict the risk of refeeding syndrome and adjust the TPN formulation accordingly. The use of AI in HPN is transforming the way patients manage their nutritional therapy outside of hospital settings. Portable devices equipped with AI algorithms can monitor patients’ metabolic status and adjust the TPN formulations in real time. However, one of the primary concerns is data privacy, as AI systems require access to sensitive patient information. Ensuring the security of this data is critical to maintaining patient trust. Another challenge is the transparency of AI algorithms. Additionally, the high cost of AI technologies may limit their accessibility in resource-constrained settings. Future research should focus on developing cost-effective AI solutions that can be implemented in a wide range of healthcare environments [37,38].

## 8. Conclusions

In recent years, parenteral nutrition has set new standards in clinical nutrition practice regarding its adaptation to specific patient requirements. The stability of the lipid fraction within PN formulations is a key element of its safety and efficacy. Understanding the mechanisms of destabilization and implementing appropriate preventive strategies can minimize the risk of complications and improve treatment outcomes. Continuous education of medical nutrition teams and conducting scientific research on the stability of TPN remain essential for the further development of this form of therapy. Despite the promising direction that parenteral nutrition has taken in recent years, there is still a lack of continuity between the results of research centers (e.g., regarding the stability of mixtures after adding drugs or the impact of packaging on the stability of the mixture) and current clinical practice. There is a lack of implementation of modern changes in underfunded clinical departments of hospitals. Strengthening the cooperation between research centers and clinical nutrition laboratories would facilitate the creation of consistent guidelines, contributing to the development of patient-targeted parenteral nutrition strategies. The popularization of home parenteral nutrition is desirable for patients, but it is still a challenge for pharmacists to prepare ready-made HPN compositions. On the other hand, the addition of vitamins to the TPN bag by the patient in non-aseptic conditions is not ideal and carries a risk of microbial contamination.

The growing diversity of commercially available TPN components has increased the risk of physicochemical incompatibilities, necessitating formulation and compatibility assessments. Compatibility issues can lead to degradation of lipid emulsions, precipitation, and other adverse outcomes, potentially endangering patients. TPN continues to evolve, and it holds the potential to transform the recovery of patients with complex nutritional needs. The modern approach to TPN reflects a comprehensive, patient-centered strategy that integrates precision in nutrition, cutting-edge technology, and multidisciplinary expertise.

The co-administration of drugs and PN presents both challenges and opportunities. While compatibility issues remain a significant concern, advances in PN formulations and compatibility testing have improved the safety of this practice. To ensure the further development of safe parenteral nutrition, it is necessary to conduct broad stability assays of new TPN compositions and focus on their interactions with the injected additives, including vitamins and drugs. The results of physicochemical stability studies should be supplemented with clinical studies on the effectiveness of administering the drugs in a TPN admixture. Each modification of the parenteral nutrition process should improve the comfort of the hospitalized patient and shorten the recovery time while limiting unnecessary painful medical procedures. The materials used for packaging PN should be carefully considered to prevent leaching during storage. The recommendations in the ESPEN and ASPEN guidelines should be updated, and they should provide strategies for enhancing safety and clinical outcomes. To ensure the safety and efficacy of this advanced nutritional therapy, continuous professional education and monitoring of research and clinical guidelines are imperative for optimizing TPN management in contemporary medical practice.

## Figures and Tables

**Figure 1 nutrients-17-00846-f001:**
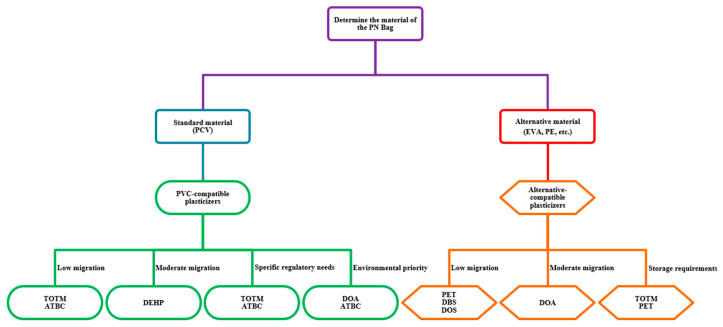
Flow chart for selecting the type of PN packaging.

**Table 1 nutrients-17-00846-t001:** Influence of co-administration of drugs on parenteral nutrition stability [9,18,19,20,21,22,23].

Drug	Compatible PN Mixtures	Incompatible PN Mixtures	Key Notes
Furosemide	Nutriflex Lipid Special^®^ Braun, Germany (immediate infusion only)	N/A	Degradation under light and at 25 °C (Not suitable for prolonged storage)
Torasemide	Nutriflex Lipid Special^®^ Braun, Germany (immediate infusion only)	N/A	Requires immediate infusion to ensure stability
Ketoprofen	Lipoflex Special^®^ Braun, Germany Omegaflex Special^®^ Braun, Germany Olimel Peri N4E^®^ Baxter, Poland	Kabiven^®^ Fresenius Kabi AB, Sweden SmofKabiven^®^ Fresenius Kabi AB, Sweden Kabiven^®^ Fresenius Kabi AB, Sweden Kabiven Peripheral^®^	Lipid instability noted in incompatible mixtures
Paracetamol	Nutriflex Lipid Special^®^ Olimel N5E^®^ Baxter, Poland Numeta G16E^®^ Baxter, Poland Kabiven^®^ Fresenius Kabi AB, Sweden SmofKabiven^®^ Fresenius Kabi AB, Sweden Kabiven^®^ Fresenius Kabi AB, Sweden	N/A	Stable in tested PN formulations under Y-site infusion conditions
Morphine Sulfate	Nutriflex Lipid Special^®^	N/A	Stable when mixed with large PN volumes (3000 mL)
Sodium Valproate	Omegaflex Special^®^ Braun, Germany Omegaflex Peri^®^ Braun, Germany	Lipoflex Special^®^ Braun, Germany Lipoflex Peri^®^ Kabiven^®^ Fresenius Kabi AB, Sweden SmofKabiven^®^ Fresenius Kabi AB, Sweden Kabiven^®^ Fresenius Kabi AB, Sweden Kabiven Peripheral^®^ Olimel Peri N4E^®^ Baxter, Poland	Lipid particle size (>1000 nm incompatible admixtures
Colistin (COL)	Kabiven^®^ Fresenius Kabi AB, Sweden SmofKabiven^®^ Fresenius Kabi AB, SwedenKabiven^®^ Fresenius Kabi AB, Sweden Olimel N9E^®^ Baxter, PolandOlimel N9E^®^ Baxter, Poland^®^ Baxter, Poland Nutriflex Lipid Special^®^ Nutriflex Omega Special^®^ Braun, Germany	N/A	COL enhances emulsion stability by reducing zeta potential

**Table 2 nutrients-17-00846-t002:** Characteristics of selected plasticizers.

Type of Plasticizer	Example	Characteristics	Advantages	Disadvantages	Applications in PN Bags
Phthalates	DEHP (Di(2-ethylhexyl) phthalate)	Traditionally used in PVC Provides high flexibility	✓ Good flexibility ✓ Resistant to mechanical damage	⌧ Potential toxicity ⌧ Migration into lipid solutions poses risks for patients, especially neonates	Rarely used due to health concerns Being replaced by safer alternatives
Trimellitates	TOTM (Tri-octyl trimellitate)	Alternative to DEHP in PVC Reduced migration into lipid solutions	✓ High chemical stability ✓ Low migration	⌧ Higher production costs ⌧ Less flexibility compared to DEHP	Used in PVC bags as a safer alternative to DEHP
Citrates	ATBC (Acetyl tributyl citrate)	Low-toxicity plasticizer based on citrates Suitable for sensitive applications	✓ Good flexibility ✓ Biodegradable	⌧ Reduced heat resistance ⌧ Possible migration into solutions	Applied in medical devices as a safe alternative to phthalates
Polymeric Plasticizers	PET Poly(ethylene terephthalate)	High-molecular weight plasticizers No migration into solutions	✓ High stability ✓ Absence of migration	⌧ High cost ⌧ More complex manufacturing process	Used in modern PN bags made of polyolefins such as EVA and PE
Adipate Esters	DOA (Dioctyl adipate)	Based on adipic acid Ensures adequate flexibility	✓ Stable in lipid solutions ✓ Low toxicity	⌧ Limited availability ⌧ Potential for some migration	Applied as an alternative to phthalates in specific medical applications

## Data Availability

No new data were created.

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
