# Peer review of "The Modern Approach to Total Parenteral Nutrition: Multidirectional Therapy Perspectives with a Focus on the Physicochemical Stability of the Lipid Fraction"

_nutrients, 2025, doi:10.3390/nu17050846_

Round 1

Reviewer 1 Report

Comments and Suggestions for Authors

This manuscript comprehensively addresses various clinical aspects of total parenteral nutrition (TPN), including physicochemical stability, drug co-administration compatibility, catheter-related complications, and supply shortages. However, as a review article, several major issues need to be addressed:

  1. The topic, abstract, and conclusion are too generalized and fail to clearly summarize the study’s key findings. The abstract should concisely highlight the main takeaways of the review rather than broadly discussing TPN trends. Key findings, such as specific compatibility issues or recommendations for clinical practice, should be explicitly stated. Similarly, the conclusion should summarize the manuscript's major contributions and practical implications rather than restating general themes.
  2. The introduction lacks a clear purpose and justification for the selected topics. The authors should explicitly state why these specific topics were chosen and their significance in modern TPN practice. Currently, the introduction does not sufficiently explain the rationale behind discussing these aspects. Adding a brief background on current challenges in TPN and how this review addresses them would improve clarity and relevance.
  3. The discussion in each section is overly lengthy and lacks focus.In the first topic, A Patient-Targeted Approach, the discussion starts with lipid emulsions but abruptly shifts to home parenteral nutrition (HPN) without a logical transition or connection. This section lacks clinical recommendations or a concluding statement, making it difficult for readers to extract key information. Similar issues exist in other parts of the article, where lengthy paragraphs lack subtitles, summaries, or practical applications. To improve readability, each section should have clear subheadings, structured discussions, and clinically relevant conclusions.
  4. The section “Co-Administration of Drugs - Parenteral Nutrition Stability and Potential Drug Degradation” provides useful information on several drugs. However, it is unclear why these specific drugs were chosen. The selection criteria for the discussed drugs should be clearly explained in the introduction or within the section itself. Were these drugs chosen based on their frequency of use in TPN, their known compatibility issues, or recent clinical concerns? Providing a rationale for drug selection would improve the manuscript’s clarity and strengthen its relevance to clinical practice.
Comments on the Quality of English Language

not very good 

Author Response

Please see the attachment, 
We would like to kindly inform you that all the suggested corrections and comments have been carefully addressed and incorporated directly into the manuscript. For your convenience, all changes have been highlighted in red.
Thank you once again for your valuable feedback, which has significantly contributed to improving the quality of the article.
Should you require any further information or clarification, please do not hesitate to contact us. We also used an MPDI edithor author's services to improve our quality of english in publication.

Reviewer 2 Report

Comments and Suggestions for Authors

The manuscript offers a comprehensive overview of modern Total Parenteral Nutrition (TPN), with a particular focus on physicochemical stability and the lipid fraction. While the paper successfully highlights critical issues related to lipid emulsions, drug compatibility, and catheter-related complications, the overall scope appears somewhat unclear. From the title and abstract, one would expect a broad discussion on TPN, yet the review primarily delves into lipid formulations. The authors should clarify their intended focus early in the manuscript to better align reader expectations with the actual content.

Additionally, while the manuscript provides an extensive discussion on various aspects of TPN, it does not specify the criteria used to select the included evidence. Even though this is a narrative review, a brief statement on the methodology—such as how relevant literature was identified and selected—would significantly improve transparency. Mentioning key databases searched, the types of studies considered, or any timeframe restrictions would provide readers with a clearer understanding of how the evidence was curated.

Another area for improvement lies in the conclusion section, which currently feels somewhat underdeveloped. Instead of merely summarizing key points, the authors should take this opportunity to synthesize their findings, highlight critical research gaps, and offer a forward-looking perspective. Expanding on unresolved challenges in TPN, particularly in relation to lipid formulations and stability issues, would enhance the manuscript’s contribution to the field.

Furthermore, the review would benefit from a stronger analytical voice from the authors. While it effectively compiles existing knowledge, there is limited discussion of controversies, unresolved questions, or potential innovations. The authors are encouraged to go beyond summarizing the literature and provide their own perspectives on the strengths and limitations of current TPN approaches. Engaging critically with the topic—perhaps by discussing areas where clinical guidelines remain ambiguous or debating the feasibility of emerging solutions—would make the review more compelling and valuable to readers.

Author Response

(The authors gave the same response as above.)

Round 2

Reviewer 1 Report

Comments and Suggestions for Authors

The manuscript have been revised and more comprehensive. It provide informaiton about the PN may intestested the readers. 

Reviewer 2 Report

Comments and Suggestions for Authors

thanks for having considered my comments